# LESS IS MORE: TOWARD ZERO-SHOT LOCAL SCENE GRAPH GENERATION VIA FOUNDATION MODELS

## ABSTRACT

Humans inherently recognize objects via selective visual perception, transform specific regions from the visual field into structured symbolic knowledge, and reason their relationships among regions based on the allocation of limited attention resources in line with humans' goals (Folk et al., 1992). While it is intuitive for humans, contemporary perception systems falter in extracting structural information due to the intricate cognitive abilities and commonsense knowledge required. To fill this gap, we present a new task called `Local Scene Graph Generation`. Distinct from the conventional scene graph generation task, which encompasses generating all objects and relationships in an image, our proposed task aims to abstract pertinent structural information with partial objects and their relationships for boosting downstream tasks that demand advanced comprehension and reasoning capabilities. Correspondingly, we introduce z**E**ro-shot **L**ocal sc**E**ne **G**r**A**ph ge**N**era**T**ion (ELEGANT), a framework harnessing foundation models renowned for their powerful perception and commonsense reasoning, where collaboration and information communication among foundation models yield superior outcomes and realize zero-shot local scene graph generation without requiring labeled supervision. Furthermore, we propose a novel open-ended evaluation metric, **E**ntity-level **CLIPS**cor**E** (ECLIPSE), surpassing previous closed-set evaluation metrics by transcending their limited label space, offering a broader assessment. Experiment results show that our approach markedly outperforms baselines in the open-ended evaluation setting, and it also achieves a significant performance boost of up to 24.58% over prior methods in the close-set setting, demonstrating the effectiveness and powerful reasoning ability of our proposed framework.

## 1 INTRODUCTION

Human visual perception is adeptly synchronized with ongoing activity, highlighting salient objects and swiftly deducing their relationships in novel contexts. Such cognitive proficiency underpins our intuitive structural information extraction. For instance, when hungry, by identifying a pizza on a plate or inside a microwave, we can easily obtain the structural information (`pizza`, `on`, `plate`) or (`pizza`, `in`, `microwave`) to facilitate subsequent decisions, e.g., getting the pizza from the plate or opening the microwave. The perceptual competence also holds promise for enhancing AI systems' comprehension and reasoning capabilities, as evidenced by advancements in visual question answering (Han et al., 2021; Jiang et al., 2020) and image captioning (Yang et al., 2022b; Zhang et al., 2021).

In this paper, we delve into scene graphs - a structured representation of visual scenes wherein entities are graph nodes connected by labeled edges denoting their relationships. Such representations bridge the chasm between raw pixels and semantic comprehension, offering valuable information for diverse computer vision tasks. Despite recent strides in scene graph generation (Zhang et al., 2023b; Jung et al., 2023; Kundu & Aakur, 2023; Zheng et al., 2023a), translating these advancements into effective scene graph generation tools still presents challenges in novel environments due to the noisy supervision, constrained label space, and long-tailed relationship distribution, hindering the direct applicability of these approaches in downstream tasks (Li et al., 2022a; Yao et al., 2021). Consequently, ground truth scene graphs are often leveraged in downstream tasks (Puig et al., 2021), narrowing their versatility.

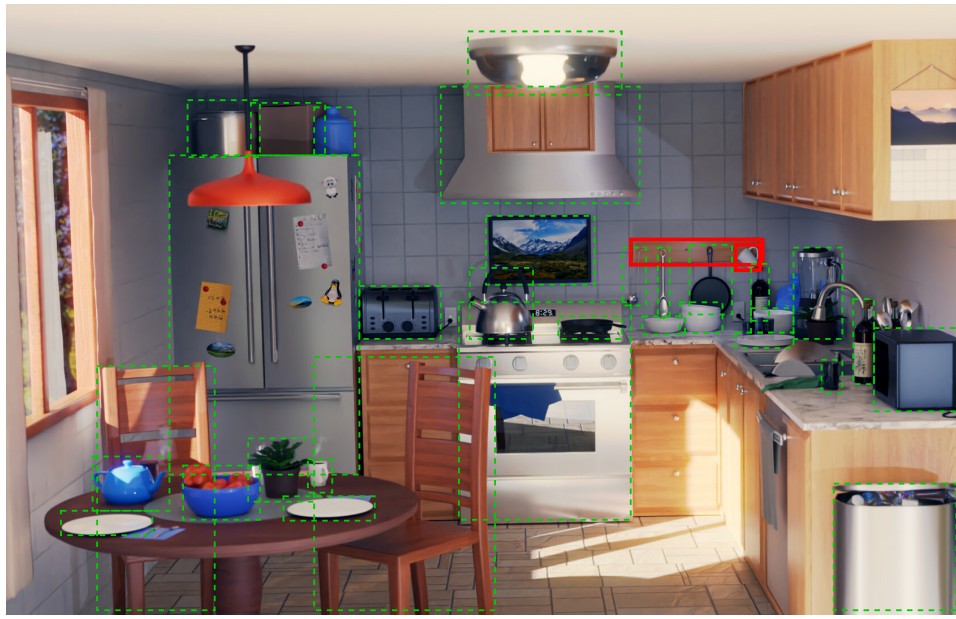

Figure 1: Local vs. Global Scene Graphs. While global scene graph methods detect all entities (represented by green dashed boxes) and their relationships, we argue that for instructions like "`obtain the white cup`," a local scene graph, exemplified by (`cup, mounted on, shelf`) (highlighted by red solid boxes), is more consistent with the human recognition process and adequate for guiding subsequent actions, such as unmounting the white cup from the shelf.

In light of these challenges, we propose emphasizing select entities and relationships aligned with specific tasks - echoing human cognition. It inspires the introduction of the "local scene graph," a departure from the conventional "global scene graph," as illustrated in Figure 1. Local scene graphs encapsulate task-pertinent entities and relationships. For instance, when performing an instruction such as "`obtain the white cup`," global scene graph generation approaches recognize all entities and relationships, inevitably introducing cumulative errors detrimental to downstream tasks from individual inaccurate relationship estimation. In contrast, a local scene graph succinctly illuminates the cup's position on a shelf, ignoring entities and relationships unrelated to the current task and streamlining the subsequent task, i.e., unmounting the white cup from the shelf, which aligns with the human cognition process (Folk et al., 1992).

Therefore, we introduce a new task named `local scene graph generation`, which generates a local scene graph according to given entities. Correspondingly, considering the scarcity of annotated scene graph data, we propose a z**E**ro-shot **L**ocal sc**E**ne **G**r**A**ph ge**N**era**T**ion (ELEGANT) framework harnessing foundation models enriched with perceptual and commonsense reasoning. It emphasizes model synergy, producing superior outcomes and realizing zero-shot local scene graph generation without labeled supervision. Specifically, given a subject selected by humans, an observer model outputs objects associated with the subject. A thinker model then identifies possible relationships, delegating them to a verifier model to validate their correctness. However, simply combining models leads to subpar results due to individual model limitations. Therefore, we advocate a **Co**-**Ca**libration (CoCa) strategy, calibrating model-specific knowledge through cross-model knowledge exchange to enhance model collaboration. Furthermore, we introduce a novel open-ended evaluation metric, **E**ntity-level **CLIPS**cor**E** (ECLIPSE), surpassing previous closed-set evaluation metrics by transcending their limited label space, offering a broader assessment.

We summarize the main contributions as follows:

- We propose a new task, local scene graph generation, to abstract pertinent structural information with partial objects and relationships, which aligns with human cognition and improves downstream tasks demanding intricate comprehension and reasoning.

- We devise a new framework, zero-shot local scene graph generation, to abstract local scene graphs without labeled supervision, exploiting foundation models renowned for their powerful perception and commonsense reasoning. Moreover, we propose a co-calibration strategy, fostering knowledge exchange to amplify model cooperation.

- We introduce a new open-ended evaluation metric, entity-level CLIPScore, to evaluate our proposed method in an open-ended setting.

## 2 RELATED WORK

**Supervised Scene Graph Generation** has attracted substantial attention within the domain of computer vision due to its pivotal role in bridging the chasm between fundamental low-level visual features and the elevated semantic comprehension of visual scenes, offering a valuable substrate for diverse computer vision tasks (Han et al., 2021; Jiang et al., 2020; Yang et al., 2022b; Zhang et al., 2021). The panorama of existing scene graph generation models encompasses both single-stage and two-stage methodologies. Drawing inspiration from the single-stage object detection model DETR (Carion et al., 2020), single-stage scene graph generation models (Liu et al., 2021; Shit et al., 2022; Li et al., 2022b; Cong et al., 2023; Teng & Wang, 2022) directly prognosticate pair proposals and effectuate scene graph synthesis through learnable queries. In contrast, two-stage methods (Kundu & Aakur, 2023; Wang et al., 2019; Shi et al., 2021; Chen et al., 2019) employ pre-trained object detectors to identify image regions, which serve as nodes in the scene graph. Subsequently, relationships between entities are delineated through relationship classification for edge labeling. To bolster the fidelity of entity representations with contextual insights, models incorporate diverse modules to facilitate the fusion of information across graph nodes and image contexts, including transformer (Vaswani et al., 2017; Li et al., 2022b; Kundu & Aakur, 2023) and graph neural network (Li et al., 2021; Khademi & Schulte, 2020). However, existing scene graph generation models generate global scene graphs and cannot directly generate local scene graphs given a subject.

**Zero-Shot Scene Graph Generation** marks an innovative direction, abstracting scene graphs without labeled supervision. (Yao et al., 2021) extracts possible (`subject`, `relationship`, `object`) triplets from a knowledge base, Conceptual Captions (Sharma et al., 2018), and exploit a CLIP model (Radford et al., 2021) for verifying the correctness of relationship candidates. (Li et al., 2023b), on the other hand, leverages large language models to furnish detailed descriptions of visual cues for relationships, subsequently deploying prompts to a CLIP model for relationship prediction. However, knowledge-base-driven triplets only capture partial relationships, and the scalability of description generation-based methods falter with the advent of new entities or relationships. Moreover, the constrained label space limits a broader assessment. In this paper, we propose a zero-shot local scene graph generation framework harnessing foundation models to extract and evaluate openvocabulary local scene graphs.

**Foundation Model Collaboration** has recently received significant attention. The success of large language models (LLMs) (Brown et al., 2020; OpenAI, 2023) leads to employing LLMs as controllers to integrate various foundation models (Bommasani et al., 2021) for multi-modal reasoning. (Zhang et al., 2023a) build a cooperative embodied agent to collaborate with other agents and humans to decompose tasks with LLMs. (Wang et al., 2023) mimic the cognitive synergy in human intelligence using a single LLM but acting in different roles to collaborate by prompting. In this paper, we leverage foundation models, renowned for their powerful perception and commonsense reasoning, to obtain local scene graphs.

## 3 LOCAL SCENE GRAPH GENERATION

### 3.1 PROBLEM DEFINITION

Given an image and a subject $s$ in this image, the local scene graph generation task needs models to abstract a local scene graph $\mathbb{G} = \{\mathbb{E}, \mathbb{R}\}$, where nodes $\mathbb{E}$ contain subject $s$ and the objects $\mathbb{O} = \{o_1, o_2, \cdots\}$ associated with $s$; edges $\mathbb{R}$ include relationships between $s$ and $o_i$. Compared with the global scene graph generation task, which generates all the entities and relationships in an image, our proposed local scene graph generation task fixes the subject as $s$ and only extracts objects and relationships associated with $s$.

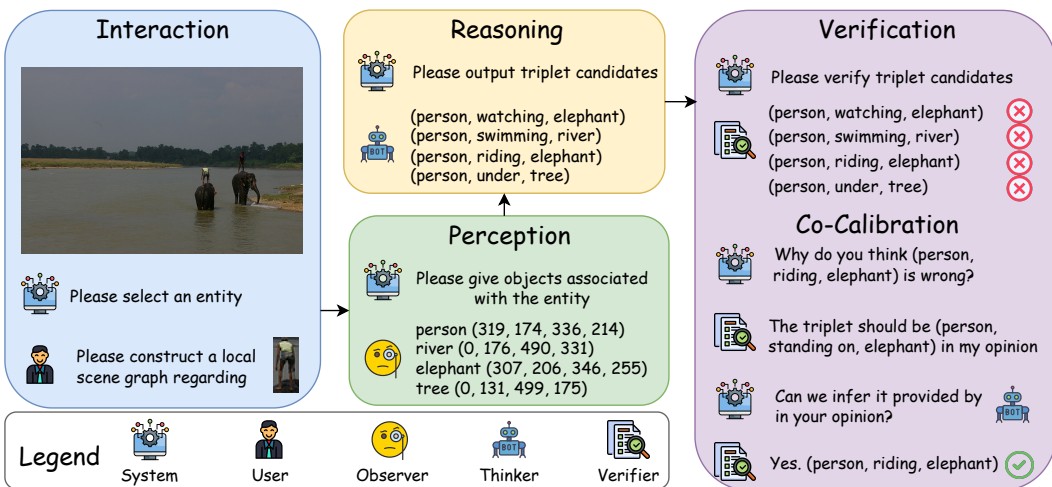

Figure 2: Pipeline of the ELEGANT Framework. For a given image and a user-specified entity as the subject, the observer identifies associated entities as objects. The thinker, equipped with robust reasoning and rich commonsense knowledge, proposes relationship candidates. Subsequent validation by the verifier ensures their relevance. The CoCa strategy then refines these results, enhancing prediction quality.

## 3.2 ZERO-SHOT LOCAL SCENE GRAPH GENERATION

We present a zEro-shot Local scEne GrAph geNeraTion (ELEGANT) framework harnessing foundation models renowned for their powerful perception and commonsense reasoning, where collaboration and information communication among foundation models yield superior outcomes and realize zero-shot local scene graph generation without requiring labeled supervision. It comprises three procedures: (1) `Perception`: Detect open-vocabulary objects related to the specified subject. (2) `Reasoning`: Deduce open-vocabulary relationships between the subject and detected objects. (3) `Verification`: Ascertain the validity of the triplets candidates. Figure 2 shows the pipeline. The subject is determined by various control signals provided by humans, including points, boxes, sentences, etc. It is worth noting that ELEGANT's modular nature allows the swapping of any internal model for its more potent counterparts to enhance performance.

**Perception.** We employ an `observer` model designed to discern open-vocabulary objects within the image. Although any suitable detection or segmentation model fits into the framework, we lean on the Segment Anything Model (SAM) (Kirillov et al., 2023), an open-vocabulary segmentation model with impressive zero-shot transferability, benefiting from promptable pre-training on a large dataset. However, SAM is a class-agnostic model, and it lacks the capability to produce symbolic semantic labels for the following reasoning stage. Recent endeavors have sought to infuse SAM with semantic information, and we utilize the GroundedSAM (Liu et al., 2023) as the observer model in this paper.

**Reasoning.** After obtaining the subject and objects, potential relationships are inferred, leveraging commonsense knowledge. GPT-like LLMs (Ouyang et al., 2022; OpenAI, 2023; Anil et al., 2023) recently emerged as powerful reasoners showcasing profound reasoning and commonsense knowledge when prompted suitably (You et al., 2023; Zhu et al., 2023). Therefore, We use an LLM as a `thinker` model and prompt it as a commonsense reasoner, subsequently yielding relationship triplet candidates. We detail the specific prompts in Appendix A.1.

**Verification.** Although LLMs have powerful commonsense knowledge, they cannot directly receive visual information, leading to inevitable bias. We introduce a `verifier` model, bridging the visual and linguistic realms, to check if these triplet candidates are correct. Direct verifying triplets, however, is fraught with challenges. For instance, posing the query, "`Does the image contain ({subject}, {relationship}, {object})?`" to the verifier yields unsatisfactory results due to its inherent model limitations. To circumvent this, we devise a structured query,

"`Question: is the {subject} {relationship} the {object}?`" and ask the verifier to answer a yes/no question, which is consistent with the pre-training tasks of the verifier and yields enhanced results. Still, the semantic divergence between models (Enser & Sandom, 2003) might result in the verifier misconstruing the thinker's commonsense knowledge, as illustrated in Figure 2. The verifier gives a wrong answer about the triplet "`(person, riding, elephant)`." To ground the knowledge in the verifier with the powerful reasoner, we introduce a `Co-Calibration` (CoCa) strategy, where the reasoner, mimicking a teacher, enables the verifier as a student to self-diagnose errors. Specifically, we ask the verifier to provide its rationale when negating a relationship. Then, we would like to know if the rationale given by the verifier can be grounded with the commonsense knowledge provided by the reasoner: "`Can we infer {A} from {B}?`" where {A} and {B} are structured triplets obtained by the reasoner and verifier, respectively. Because LLMs cannot access visual information directly, it may lead to bias. If the answer is still `no`, the triplet is probably wrong, and we discard the triplet. If the answer is `yes`, the knowledge is calibrated, and we keep the triplet. The detailed prompts are listed in Appendix A.1.

### 3.3 Open-Ended Evaluation Metric

Our proposed framework has the ability to predict open-vocabulary objects and relationships. However, vanilla scene graph evaluation metrics, confined to a fixed label space, tend to overlook or exclude entities and relationships outside the prescribed vocabulary, hampering a comprehensive scene graph evaluation. Recently, CLIPScore (Hessel et al., 2021) emerged as a promising open-ended evaluation technique, harnessing the CLIP model to offer a robust automatic evaluation for image captioning tasks. CLIPScore encodes both image and text features and calculates a similarity score between two modalities to represent their relevance:

$$\text{CLIPScore}(\mathbf{I}, \mathbf{C}) = \max(100 * \cos(\text{E}_{\mathbf{I}}, \text{E}_{\mathbf{C}}), 0), \tag{1}$$

where $\mathbf{I}$ is the image; $\mathbf{C}$ is the caption; $\text{E}_{\mathbf{I}}$ is the image feature extracted from the vision encoder of the CLIP model; $\text{E}_{\mathbf{C}}$ is the text feature extracted from the text encoder of the CLIP model; $\cos(\cdot)$ is the cosine similarity.

To tailor this evaluation metric for local scene graph generation, for each triplet $\mathbf{C}_i = (s, r_i, o_i)$, where $r_i \in \mathbb{R}$ and $o_i \in \mathbb{O}$ within a local scene graph $\mathbb{G}$, we obscure the background to derive a masked image $\mathbf{I}_i$ based on bounding boxes of $s$ and $o_i$. Meanwhile, we rewrite the triplet as "`The {s} is {r_i} the {o_i}`." Subsequently, the CLIPScore for the triplet is computed as $\text{CLIPScore}(\mathbf{I}_i, \mathbf{C}_i)$. By calculating and averaging the CLIPScores of all triplets in $\mathbb{G}$, we derive the cumulative CLIPScore for $\mathbb{G}$:

$$\text{CLIPScore}(\mathbb{G}) = \frac{1}{|\mathbb{G}|} \sum_{i=1}^{|\mathbb{G}|} \text{CLIPScore}(\mathbf{I}_i, \mathbf{C}_i), \tag{2}$$

where $|\mathbb{G}|$ is the number of triplets in $\mathbb{G}$.

While CLIPScore offers evaluation in the open-ended setting, its original design caters to image-level evaluations, ignoring the importance of prediction length in our task. For instance, model A gives a prediction with the highest confidence and might achieve a superior score over model B, which offers three predictions. Additionally, cases where a model produces synonymous relationships for a relationship `inside`, such as `in` and `inner`, might attain high CLIPScores but fail to offer novel semantic insight (Li et al., 2022a).

In light of this, we craft a penalty function, which targets predictions that are either overly brief or elongated, with a heightened penalty for the former, advocating the generation of richer triplets. While numerous function forms are plausible, we initiate our exploration with the log barrier function (Murray & Wright, 1994):

$$y = x - \mu \log(x - 1), \tag{3}$$

where $\mu > 0$ is a scalar.

Because the log barrier function is a convex function, the minimum value is $y^* = \mu - \mu \log(\mu) + 1$ at $x = \mu + 1$. Then, we adjust equation 3 places this minimal value at 0:

$$y^{'} = y - y^*. \tag{4}$$

Given the range of $y^{'}$ is $[0, +\infty]$, we modify the range to [0, 1] by integrating $\exp(-x)$ and an $\alpha$ parameter to control the penalty strength for enhancing the flexibility of the evaluation metric:

$$y^{''} = \exp(-\alpha y^{'}). \tag{5}$$

Nevertheless, there is still a challenge: the variability in penalty scores across diverse prediction lengths makes it difficult to evaluate models fairly (Papineni et al., 2002). Therefore, we compute the average prediction length $m^*$ across the dataset and set $\mu = m^* - 1$. The penalty function is:

$$P(x) = \exp\left(-\alpha\left(x + (m^* - 1)\log\frac{m^* - 1}{x - 1} - m^*\right)\right), \tag{6}$$

where $x$ is the length of prediction. The image of the function is shown in Appendix A.2.

Finally, we present our novel open-ended evaluation metric called **E**ntity-level **CLIPS**cor**E** (ECLIPSE):

$$\text{ECLIPSE}(\mathbb{G}) = P(|\mathbb{G}|)\frac{1}{|\mathbb{G}|}\sum_{i=1}^{|\mathbb{G}|}\text{CLIPScore}(\mathbf{I}_i, \mathbf{C}_i), \tag{7}$$

where $-\frac{d\,P}{d|\mathbb{G}|}\Big|_{|\mathbb{G}|\to 1} > -\frac{d\,P}{d|\mathbb{G}|}\Big|_{|\mathbb{G}|\to+\infty}$. Therefore, shorter predictions receive a larger penalty than longer predictions.

## 4 EXPERIMENTS

### 4.1 EXPERIMENT SETUP

**Datasets.** To evaluate our method, we utilize 1) Visual Genome (Krishna et al., 2017), which contains $26,443$ images for testing, and each image is manually annotated with entities and relationships. 2) GQA (Hudson & Manning, 2019) containing $8,208$ images for testing, whose split provided by (Li et al., 2023b). To ensure consistent benchmarking against prior zero-shot scene graph generation works (Yao et al., 2021; Li et al., 2023b), we also conduct experiments within a closed-set paradigm on the Visual Genome dataset, where (Yao et al., 2021) removes hyperyms and redundant synonyms in the most frequent 50 relation categories, resulting in 20 well-defined relation categories, and (Li et al., 2023b) adopts the 24 semantic relationship classes.

**Evaluation Metrics.** We iteratively select one from all entities in an image as the subject and generate a local scene graph to construct a global scene graph for evaluation. We conduct experiments on both open-ended and closed-set settings. For the open-ended setting, our proposed ECLIPSE is reported. For the closed-set setting, we report Recall@K (R@K), which indicates the proportion of ground truths that appear among the top-K confident predictions, and Mean Recall@K (mR@K), which averages R@K for each category.

**Implementation Details.** Our observer model is the GroundedSAM (Liu et al., 2023), an open-vocabulary segmentation model with semantic information. The thinker model is based on the GPT-3.5-turbo (OpenAI, 2023), a large language model with impressive reasoning skills and commonsense knowledge. For the verifier model, we deploy the BLIP2 (Li et al., 2023a), a pre-trained vision-language model.

### 4.2 OPEN-ENDED LOCAL SCENE GRAPH GENERATION EVALUATION

We introduce diverse baselines to evaluate the efficiency of our proposed framework. Table 1 illustrates the results on the Visual Genome dataset. The results on the GQA dataset are shown in Appendix A.3.

**Observer.** We employ open-vocabulary (GroundedSAM (Liu et al., 2023)) and closed-set detectors (FasterRCNN (Ren et al., 2015), which were widely utilized in previous approaches) to demonstrate the effectiveness of perception performance. GroundedSAM markedly outperforms FasterRCNN, attributed to its open-vocabulary perception capabilities, whereas FasterRCNN can only recognize objects defined in a fixed label space. A comparative analysis reveals that our method

Table 1: Open-ended evaluation results on the test set of the Visual Genome dataset. The parameter $\alpha$ is set as $0.01$.

| Model | | | ECLIPSE |
| --- | --- | --- | --- |
| Observer | Thinker | Verifier | |
| Faster RCNN | GPT-3.5-Turbo | BLIP2 OPT 6.7B | 19.31 |
| GroundedSAM | OPT 2.7B | BLIP2 OPT 6.7B | 0.09 |
| GroundedSAM | OPT 6.7B | BLIP2 OPT 6.7B | 0.16 |
| GroundedSAM | LLaMA2 7B | BLIP2 OPT 6.7B | 19.01 |
| GroundedSAM | Vicuna 7B | BLIP2 OPT 6.7B | 20.41 |
| GroundedSAM | GPT-3.5-Turbo | BLIP2 FlanT5 XL | 20.97 |
| GroundedSAM | GPT-3.5-Turbo | BLIP2 FlanT5 XXL | 21.20 |
| GroundedSAM | GPT-3.5-Turbo | BLIP2 OPT 2.7B | 21.50 |
| **GroundedSAM** | **GPT-3.5-Turbo** | **BLIP2 OPT 6.7B** | **21.54** |

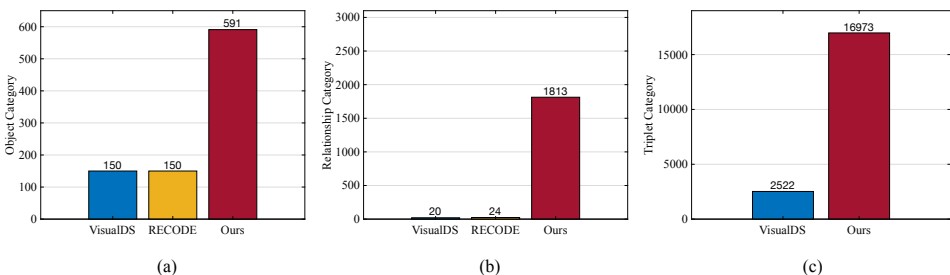

Figure 3: Assessing Prediction Diversity. (a) Number of entity categories. (b) Number of relationship categories. (c) Number of predicted triplets.

identifies approximately 4x more object categories than previous closed-set methods, as illustrated in Figure 3 (a).

**Thinker.** The results show that the OPT models Zhang et al. (2022) do not perform well due to limited reasoning ability. In comparison, LLaMA2 (Touvron et al., 2023) and Vicuna (Zheng et al., 2023b) show comparative performance due to Instruction Tuning (Longpre et al., 2023). GPT-3.5-Turbo (OpenAI, 2023) exhibits superior performance, attributable to its enhanced reasoning capabilities. Compared with closed-set methods, our method produces about 25x more relationship categories, as shown in Figure 3 (b).

**Verifier.** As the verifier to verify relationship candidates, we explore various variants of BLIP2, a visual question answering model trained on large datasets. BLIP2 OPT is based on the unsupervised-trained OPT model family (Zhang et al., 2022) for decoder-based LLMs, and BLIP2 FlanT5 is based on the instruction-trained FlanT5 model family (Zhang et al., 2022) for encoder-decoder-based LLMs. The results show that BLIP2 OPT achieves higher performance than BLIP2 FlanT5. From Figure 3 (c), compared with VisualDS, our approach achieves around 7x more triplet categories.

### 4.3 COMPARISION WITH CLOSED-SET ZERO-SHOT SCENE GRAPH GENERATION METHODS

To assess the commonsense reasoning capability of our proposed method, we benchmark it against two zero-shot scene graph generation approaches: VisualDS (Yao et al., 2021) and RECODE (Li et al., 2023b). VisualDS crafts scene graphs by mining relationship candidates from knowledge bases, subsequently validated via the CLIP model (Radford et al., 2021). Furthermore, they employ the predicted scene graphs as pseudo labels for the supervised scene graph generation model training. Our comparison focuses on the first phase of VisualDS, and it's worth noting that the scene graphs we generate can similarly be harnessed for supervised training. For a fair comparison, we adopt

Table 2: Closed-set evaluation results on the test set of the Visual Genome dataset. Note the differences in the experimental setup between VisualDS and RECODE concerning the number of relationship categories (#Rel).

| Method | #Rel | R@10 | R@20 | R@50 | mR@10 | mR@20 | mR@50 |
|---|---|---|---|---|---|---|---|
| VisualDS Yao et al. (2021) | 20 | 27.72 | 33.22 | 38.21 | 16.32 | 20.49 | 24.94 |
| Ours | | **30.27** | **36.80** | **41.04** | **21.21** | **26.11** | **29.78** |
| RECODE Li et al. (2023b) | 24 | - | 10.60 | 18.30 | - | 10.70 | 18.70 |
| Ours | | **28.14** | **35.18** | **38.87** | **39.51** | **16.54** | **21.39** |

Table 3: The effectiveness of CoCa strategy on the test set of the Visual Genome dataset. E@$*$ denotes ECLIPSE when the parameter $\alpha$ is set as $*$.

| Method | R@10 | R@20 | mR@10 | mR@20 | E@0.1 | E@0.01 | E@0.001 | #Triplets |
|---|---|---|---|---|---|---|---|---|
| Ours | **30.27** | **36.80** | **21.21** | **26.11** | **17.63** | **20.39** | **20.78** | **12120** |
| - Co-Calibration | 23.64 | 31.96 | 18.78 | 25.89 | 14.86 | 16.51 | 16.73 | 7108 |

the ground truth object detections, control the generation of relationship candidates by prompts, and filter out relationships that do not exist in relationship label space. As our approach is a local scene graph generation framework, we iteratively run our method on all objects to obtain a global scene graph. The results are shown in Table 2.

From Table 2, our approach significantly improves performance up to $24.58$ in Recall. In contrast to VisualDS, which sources relationship candidates from a static knowledge base, our method leverages the vast commonsense reasoning of large language models (LLMs) pretrained on expansive datasets. Furthermore, the in-context learning ability (Wei et al., 2022) provides a powerful and effective way to receive context about the world. Prompts encompassing entities like "`cup, oven, stove, refrigerator`" hint at a kitchen scene, and LLMs might consequently produce scene-specific relationships. Meanwhile, RECODE employs LLMs to generate detailed descriptions for relationship triplets. However, it still needs to pre-define a relationship label space and generate a description for each triplet, which is time-consuming and cannot easily scale up for new entities and relationships. Our approach utilizes LLMs to generate relationship candidates, which is efficient and can effortlessly deal with new environments.

### 4.4 EFFECTIVENESS OF COCA STRATEGY

To demonstrate the effectiveness of our proposed CoCa strategy, we compare our approach with a variant devoid of the CoCa strategy. The results are shown in Table 3. In the absence of the CoCa strategy, there's a marked decline in performance, coupled with a significant reduction in predicted triplet counts, indicating that $5012$ triplets are initially recognized as negative samples by the verifier and rectified by the CoCa strategy.

### 4.5 QUALITATIVE RESULTS

Figure 4 shows the qualitative results. The red dashed box is the subject, and the green solid boxes denote objects. The results demonstrate the effectiveness of our proposed method. The first example is evaluated in the open-vocabulary setting, and other images are evaluated in the closed-set setting. The open-vocabulary detector can generate various multi-grained entities, e.g., hat, pant, shoe, and the closed-set detector can only give coarse-grained entities, e.g., child. Consequently, given more diverse entities, our method can produce a more significant number of triplets.

### 4.6 LOCAL SCENE GRAPHS FOR DOWNSTREAM TASKS

To assess the utility of local scene graphs, we incorporate them into the Visual Question Answering task. We evaluate our approach on the GQA testdev set (Hudson & Manning, 2019), selecting a

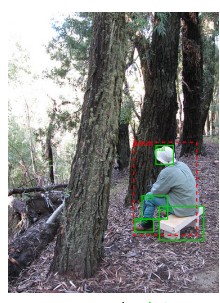 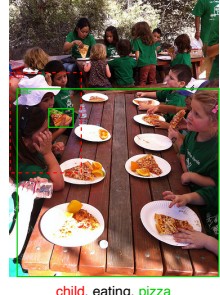 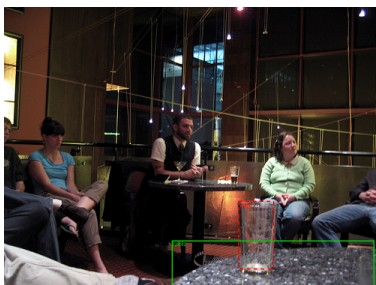

person, wearing, hat
person, wearing, pant
person, wearing, shoe
person, sitting on, chair

child, eating, pizza
child, sitting on, table

cup, standing on, table

Figure 4: Visualization of Local Scene Graphs Generation. Subjects are depicted with red text and dashed boxes, while objects are highlighted with green text and solid boxes. Relationships are denoted in black texts.

Table 4: Effectiveness of local scene graphs in VQA tasks.

| Method | Thinker | Scene Graph | ACC |
|---|---|---|---|
| Baseline | - | - | 50.4 |
| Ours | Vicuna | global | 51.9 |
| | | local | **55.4** |
| | GPT-3.5-Turbo | global | 54.2 |
| | | local | **58.3** |

random subset of $1,000$ samples. To derive local scene graphs pertinent to a given query, spaCy[1] is employed to extract nouns from the question, and the nouns then serve as subjects for creating local scene graphs via ELEGANT. Our experiments leverage the BLIP2 FlanT5 XL model (Li et al., 2023a). For a given local scene graph $\mathbb{G}$, we utilize the template "`Context: {G}. Question: {Q}, Short Answer:`" as the prompt, where `Q` is the question; `G` consists of triplets denoted as "`A {s} is {r_i} a {o_i}`." For comparison, global scene graphs are also produced. Notably, scene graphs are integrated directly into the BLIP2 model via prompting. So, we do not train or fine-tune the BLIP2 model. We report the accuracy and the results are shown in Table 4.

From Table 4, both local and global scene graphs enhance the VQA task's performance, underscoring the value of scene graphs for downstream tasks that demand intricate reasoning and commonsense knowledge. The results also suggest that local scene graphs, crafted in a task-specific manner, can offer richer commonsense knowledge insights. Moreover, the results demonstrate that models with superior reasoning capabilities yield improved results.

## 5 CONCLUSION

We introduce a novel task, Local Scene Graph Generation, to effectively abstract relevant structural information from partial objects. This task narrows the gap between human cognitive processes and AI perception systems. To address this, we present a scalable framework, termed ELEGANT, which leverages collaboration among foundational models to realize zero-shot local scene graph generation. Additionally, We propose a new open-ended evaluation metric, ECLIPSE, surpassing the limitations of previously closed-set metrics with limited label space. Experiment results show that our approach outperforms baselines in the open-ended evaluation setting and performs significantly over prior methods in the close-set setting. Moreover, the predicted local scene graphs can significantly improve intricate comprehension and reasoning in downstream tasks.

---

[1]https://spacy.io/

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

# A  APPENDIX

## A.1  PROMPTS

In this section, we present prompts utilized in this paper.

**Open-Vocabulary Thinker Prompt**

To prompt an LLM as a commonsense reasoner, we provide the following prompt:

```
You are an AI assistant with rich visual commonsense knowledge
and strong reasoning abilities.
You will be provided with:
1. A subject.
2. A list of entities.

To effectively analyze the image, you should give all possible
triplets formulated as (subject, predicate, object), where the
object is selected from entities. The predicate connecting the
subject and object should be reasonable and commonsense. A
valid triplet should be satisfied: if we rewrite (subject,
predicate, object) to "subject is predicate object," the
sentence must be reasonable. For example, (man, sitting on,
chair) is rewritten to "man is sitting on chair," and it is
reasonable. But (man, parked on, chair) is invalid because
"man is parked on chair" is unreasonable.

Subject: man
Entities: man, woman, balloon, tree
Triplets: (man, watching, woman), (man, carrying, balloon), (man,
talking to, man)
```

**Closed-Set Thinker Prompt**

For comparison with VisualDS, we provide the 20 relationship candidates:

```
You are an AI assistant with rich visual commonsense knowledge and
strong reasoning abilities.
You will be provided with:
1. A subject.
2. A list of entities.
3. A list of relationship candidates: carrying, covered in,
covering, eating, flying in, growing on, hanging from, lying on,
mounted on, painted on, parked on, playing, riding, says,
sitting on, standing on, using, walking in, walking on, watching.

To effectively analyze the image, you should give all possible
triplets formulated as (subject, predicate, object), where the
object is selected from entities and the predicate is selected
from relationship candidates. The predicate connecting the subject
and object should be reasonable and commonsense. For example,
(man, sitting on, chair) is reasonable, but (man, parked on,
chair) is unreasonable.

Subject: man
Entities: woman, balloon, balloon, tree, tree
Triplets: (man, watching, woman), (man, carrying, balloon)
```

To compare with RECODE, we provide the 24 relationship candidates:

```
You are an AI assistant with rich visual commonsense knowledge and
```

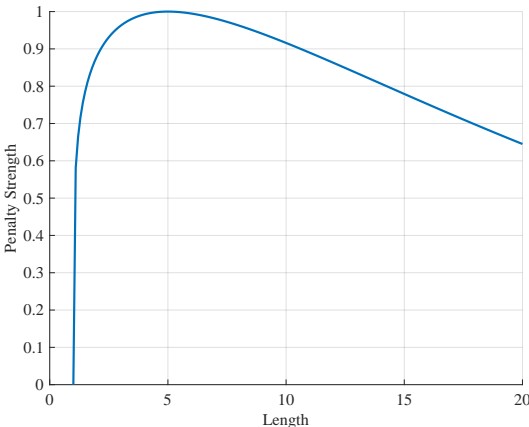

Figure 5: The image of the penalty function $P(x)$ in equation 6.

```
strong reasoning abilities.
You will be provided with:
1. A subject.
2. A list of entities.
3. A list of relationship candidates: carrying, covered in,
covering, eating, flying in, growing on, hanging from, holding,
laying on, looking at, lying on, mounted on, painted on,
parked on, playing, riding, says, sitting on, standing on, to,
using, walking in, walking on, watching.

To effectively analyze the image, you should give all possible
triplets formulated as (subject, predicate, object), where the
object is selected from entities and the predicate is selected
from relationship candidates. The predicate connecting the subject
and object should be reasonable and commonsense. For example,
(man, sitting on, chair) is reasonable, but (man, parked on,
chair) is unreasonable.

Subject: man
Entities: woman, balloon, balloon, tree, tree
Triplets: (man, watching, woman), (man, carrying, balloon)
```

**Verifier Prompt**

To verify the correctness of a triplet candidate, we use the following prompt template:

```
Question: is the {subject} {relationship} the {object}? Short
answer:
```

We present the prompt to obtain the rationale of the verifier if the answer is `no`:

```
Question: What is the relationship between {subject} and
{object}. Short Answer:
```

The following prompt is utilized to calibrate the knowledge in the verifier:

```
Context: {rationale}. Question: Can we infer "{subject} is
{relationship} {object}" from the Context? Short Answer:
```

Table 5: Open-ended evaluation results on the test set of the GQA dataset. The parameter $\alpha$ is set as 0.01.

| Model | | | ECLIPSE |
|---|---|---|---|
| Observer | Thinker | Verifier | |
| GroundedSAM | GPT-3.5-Turbo | BLIP2 OPT 2.7B | 20.68 |
| GroundedSAM | GPT-3.5-Turbo | BLIP2 OPT 6.7B | 20.82 |
| GroundedSAM | GPT-3.5-Turbo | BLIP2 FlanT5 XL | 21.19 |
| **GroundedSAM** | **GPT-3.5-Turbo** | **BLIP2 FlanT5 XXL** | **21.32** |

## A.2 THE IMAGE OF THE PENALTY FUNCTION

Figure 5 illustrates the image of the penalty function in equation 6. Because $-\frac{\mathrm{d}\,\mathrm{P}}{\mathrm{d}|\mathbb{G}|}\Big|_{|\mathbb{G}|\to 1} > -\frac{\mathrm{d}\,\mathrm{P}}{\mathrm{d}|\mathbb{G}|}\Big|_{|\mathbb{G}|\to +\infty}$, shorter predictions receive a larger penalty than longer predictions. Therefore, we penalize predictions that are either overly brief or elongated, with a heightened penalty for the former, advocating the generation of richer triplets.

## A.3 RESULTS ON THE GQA DATASET

Table 5 illustrates the results of the GQA dataset. Benefiting from the powerful reasoning ability of large language models, the generated triplets also achieve good performance. We will explore the impact of more available foundation models in the future.

## A.4 LIMITATIONS

Although we have witnessed the strong performance of our proposed method, we still have some limitations. First, our proposed framework relies on foundation models, which are time-consuming. BHowever, we can employ the framework as a pseudo-label generator to improve the performance of supervised scene graph generation methods. Second, we only deduce a relationship between a subject and an object when their IoU is larger than $0$, considering the time cost, following previous work (Yao et al., 2021). Third, the GroundedSAM can generate segmentation masks, but we only utilize the bounding boxes in this work. In the future, we can extend our framework to a zero-shot panoptic scene graph generation framework (Yang et al., 2022a).

