# OpenReview forum: "Less is More: Toward Zero-Shot Local Scene Graph Generation via Foundation Models"
_ICLR.cc/2024/Conference — ICLR 2024 Conference Withdrawn Submission_

### Official Review · Reviewer_1PLg · 2023-10-30

**Soundness:** 2 fair
**Presentation:** 2 fair
**Contribution:** 2 fair
**Rating:** 3
**Confidence:** 3

**Summary:**

This paper sets out to study local scene graph generation, aiming to align with human cognition. Authors exploit several foundation models to perform the task in a zero-shot open-vocabulary fashion.

**Strengths:**

- The motivation of this paper is interesting: it aims to align local scene graph generation (SGG) with human perception. (Please also refer to weakness.1,2).

- The proposed zero-shot system is reported to work smoothly and effectively with foundation models in both open-vocabulary and closed-set SGG.

**Weaknesses:**

- The "Segmenting anything" nature of observers conflicts with the motivation of aligning humans' selective perception, which seems self-contradictory. Humans would not typically segment anything first when given a scene image. It would be helpful if the authors could clarify this point.

- The motivation of this paper is interesting yet tricky: why can't we retrieve the local scene graph from the global scene graph? What is the advantage of directly generating a local scene graph? If the aim is to align with human perception, why is the foveated view not applied to the input scene?

- I found the novelty in this work is limited: local SGG task is derived from conventional global SGG task and the proposed system seems like a trivial effort to combine different foundation models.

- Lack of comparison: The new task and framework proposed in this paper do not appear to be exclusive. There is a possibility that such a framework could be used for what is referred to as "global SGG," or that models designed for global SGG could also perform this local SGG task. The former is reported in Table 2. I am curious about how well other global open-vocabulary SGG models perform the local SGG task [1-2].

- The analysis of experiments lacks depth. There are not many key takeaways beyond performance metrics. Additionally, figures 3-4 are less informative since open-vocabulary SGG, of course, generates more diverse triplets.

[1] Towards open-vocabulary scene graph generation with prompt-based finetuning, He et al, 2022.

[2] Learning to Generate Language-supervised and Open-vocabulary Scene Graph using Pre-trained Visual-Semantic Space, Zhang et al, 2023.

**Questions:**

Please see weaknesses above.

---

### Official Review · Reviewer_hGSC · 2023-10-30

**Soundness:** 3 good
**Presentation:** 3 good
**Contribution:** 1 poor
**Rating:** 3
**Confidence:** 4

**Summary:**

This paper proposes a prompt-based novel scene graph generation task denoted as 'local scene graph generation'. The proposed method leverages the existing foundational models to detect objects, infer relationships, and validate the inference. In addition to that, the paper proposes an evaluation metric for open-vocabulary evaluation of scene graphs. The models are evaluated on the Visual Genome and GQA dataset and compared with two existing works.

**Strengths:**

The paper has the following strengths

**1. Well-written.** The paper is easy to follow in terms of understanding their motivation and goals.

**2. Connecting Foundational model to scene graphs.** Scene Graphs have traditionally been tested on in-dataset samples such as Visual Genome or GQA. With the rise of foundational models, the open-vocabulary potential of scene graphs have risen and this paper is one of the foremost approaches in that direction.

**3. New task of scene graph generation.** Scene graphs are complex-natured entities compared to low-level object detection. Therefore, different tasks involving scene graph generation would help different researchers to concentrate on different parts. This paper's novel task creates one of such doors of exploration. Agreeing with the authors, I strongly believe the local scene graphs should be emphasized in future research since humans often look at a single entity and would like to reason its connections to other entities.

**Weaknesses:**

The main weakness of the paper is its lack of technical contribution. The paper proposes a new prompt-based task and its evaluation strategies. However, in terms of how to model the complex open-vocabulary relationship, this paper wholly relies on foundational models and report their performances. In Table 1, observer, thinker, and verifier models are borrowed from the already established model. Therefore, the claim of 'Ours' in Table 2 and Table 3 is significantly weak. This whole reliance on the off-the-shelf models makes this paper a well-written technical report of a good engineering project, but not a novel research paper.

**Questions:**

I admire the new task and evaluation of scene graph generation. However, I suggest the authors add contributions in terms of the model and revise their paper accordingly.

---

### Official Review · Reviewer_HdXh · 2023-10-31

**Soundness:** 3 good
**Presentation:** 3 good
**Contribution:** 3 good
**Rating:** 6
**Confidence:** 5

**Summary:**

The authors introduce a new task called Local Scene Graph Generation (LSGG), which aims to abstract pertinent structural information with partial objects and their relationships. To tackle this challenging task, the authors propose a zero-shot framework called ELEGANT, which harnesses foundation models for their powerful perception and commonsense reasoning. The authors evaluate their approach on a new open-ended evaluation metric called Entity-level CLIPScorE (ECLIPSE). ELEGANT achieves superior performance in both open-ended and close-set evaluation settings, demonstrating its effectiveness and powerful reasoning ability.

**Strengths:**

- Combining the scene graph data structure with the currently popular foundation models is a promising direction for the SGG field.

**Weaknesses:**

- Since ECLIPSE contains a penalty on the length of predictions, why not force the language models to output longer sentences?
- typo:
  - In A.4 LIMITATIONS, "BHowever"

**Questions:**

- As authors use other visual-linguistic models like Blip2 as a part of the whole system, why use CLIP model for evaluation?
- Are there any differences between CoCa and Chain of Thought?

---

### Official Review · Reviewer_d1KX · 2023-11-06

**Soundness:** 2 fair
**Presentation:** 3 good
**Contribution:** 2 fair
**Rating:** 5
**Confidence:** 3

**Summary:**

This paper proposes a method on zero-shot local scene graph generation using foundation models. They also proposes a new metric, 'Entity-level Clipscore' for evaluating such zero-shot scene graph generation. The authors conducted their experiments on Visual Genome and GQA dataset and reported their results using Recal@K, mR@K and their proposed metric 'ECLIPSE'

**Strengths:**

1. The paper is well-written and well-organized
2. Using foundation models and vision-language models for zero-shot or open-world exploration of scene graphs appears to be a promising approach

**Weaknesses:**

1. Since the subject line is given as an input while constructing the scene grpah, isn't it like providing the mdoel with more information considering the setting for scene graph generation(SGDet) task. The problem with locally scene graph generation is that
it has to take the help with subject line, which gives alteast one object information to start with. So the comparisons will not be fair.
2. Might get better insight if you could compare with more baselines/ aprroaches by both ECLIPSE metrics and R@K and mR@K metrics

3. In Table 3, in the third row, '- Co-Calibration' means without Co-Calibration? This might make some confusion among readers. My suggestion would be to use something like 'w/o Co-Calibration'

**Questions:**

Please explain point 1 in weakness section